# Impact of SARS-CoV-2 on healthcare and essential workers: A longitudinal study of PROMIS-29 outcomes

Jocelyn Dorney[1,2,☯], Imtiaz Ebna Mannan[2☯], Caitlin Malicki[3☯], Lauren E. Wisk[4,5☯], Joann Elmore[4,5], Kelli N. O'Laughlin[6], Dana Morse[6], Kristyn Gatling[7], Michael Gottlieb[8], Michelle Santangelo[8], Michelle L'Hommedieu[4,9], Nicole L. Gentile[10], Sharon Saydah[11], Mandy J. Hill[12], Ryan Huebinger[12], Katherine Riley Martin[13], Ahamed H. Idris[13], Efrat Kean[14,15], Kevin Schaeffer[14,15], Robert M. Rodriguez[9], Robert A. Weinstein[16,17‡], Erica S. Spatz[1,2‡*], for the INSPIRE Group[¶]

1 Section of Cardiovascular Medicine, Yale School of Medicine, New Haven, Connecticut, United States of America, 2 Center for Outcomes Research and Evaluation (CORE), New Haven, Connecticut, United States of America, 3 Department of Emergency Medicine, Yale School of Medicine, New Haven, Connecticut, United States of America, 4 Division of General Internal Medicine and Health Services Research, David Geffen School of Medicine at the University of California, Los Angeles, Los Angeles, United States of America, 5 Department of Health Policy and Management, Fielding School of Public Health at the University of California, Los Angeles, Los Angeles, United States of America, 6 Departments of Emergency Medicine and Global Health, University of Washington, Seattle, Washington, United States of America, 7 Division of Infectious Diseases, Department of Internal Medicine, Rush University Medical Center, Chicago, Illinois, United States of America, 8 Department of Emergency Medicine, Rush University Medical Center, Chicago, Illinois, United States of America, 9 Department of Medicine, University of California Riverside School of Medicine, Riverside, California, United States of America, 10 Department of Family Medicine, Department of Laboratory Medicine and Pathology, Department of Rehabilitation Medicine, Post-COVID Rehabilitation and Recovery Clinic, University of Washington, Seattle, Washington, United States of America, 11 National Center for Immunization and Respiratory Diseases, Centers for Disease Control and Prevention, Atlanta, Georgia, United States of America, 12 Department of Emergency Medicine, McGovern Medical School, UTHealth Houston, Houston, Texas, United States of America, 13 Department of Emergency Medicine, University of Texas Southwestern Medical Center, Dallas, Texas, United States of America, 14 Department of Emergency Medicine, Sidney Kimmel Medical College, Thomas Jefferson University, Philadelphia, Pennsylvania, United States of America, 15 Center for Connected Care, Thomas Jefferson University, Philadelphia, Pennsylvania, United States of America, 16 Division of Infectious Diseases, Cook County Hospital, Chicago, Illinois, United States of America, 17 Division of Infectious Diseases, Department of Internal Medicine, Rush University Medical Center, Chicago, Illinois, United States of America

☯ These authors contributed equally to this work.
‡ These authors are joint senior authors on this work.
¶ Membership of the INSPIRE Group is provided in Appendix 1.
* erica.spatz@yale.edu

## Abstract

### Importance

The mandatory service of essential workers during the COVID-19 pandemic was associated with high job stress, increased SARS-CoV-2 exposure, and limited time for recovery following infection. Understanding outcomes for frontline workers can inform planning for future pandemics.

---

**Data availability statement:** All relevant data are within the manuscript and its Supporting Information files.

**Funding:** Nicole L. Gentile MD, PhD reports receiving grants from the Agency for Healthcare Research and Quality (AHRQ U18 HS29905-01), National Center for Complementary and Integrative Health (R34AT012679-01), and National Institute on Aging (U19 AG076581-01A1) during the conduct of the study and being the primary care medical director of the Long COVID Clinic at the University of Washington which is supported by the AHRQ U18 HS29905-01 grant from the Agency for Healthcare Research and Quality. Robert M. Rodriguez reports being funded by PROCOVAXED, funded by NIAID R01AI166967-01 (PI: Rodriguez) and a global health grant from Pfizer, Inc. Michael Gottlieb reports grant funding from the Centers for Disease Control and Prevention, National Institutes of Health, Bill & Melinda Gates Foundation, GE Healthcare, Society for Academic Emergency Medicine Foundation, and the Charles J. and Margaret Roberts Fund. A partner from the CDC (Sharon Saydah, PhD, MHS) assisted with study design and provided critical feedback on this manuscript. The Innovative Support for Patients with SARS-COV-2 Infections Registry (INSPIRE) is funded by the Centers for Disease Control and Prevention (CDC, https://nam12.safelinks.protection.outlook.com/?url=http%3A%2F%2Fwww.cdc.gov-%2F&data=05%7C02%7Cerica.spatz%40yale.edu%7C41af7a32358c41302e6808ddb-3d7ec25%7Cdd8cbebb21394df8b4114e3e87abeb5c%7C0%7C0%7C638864462159780328%-7CUnknown%7CTWFpbGZsb3d8eyJFbXB0eU1hcGkOnRydWUsIlYiOiIwLjAuMDAwMCIsIlAiOiJXaW4zMiIsIkFOIjoiTWFpbCIsIldUIjoyfQ%3D%3D%7C0%7C%7C%7C&sdata=%2FHr35uh7kbkNgZPAONc1ZpXek3%2BzwUWjzugzRYBgeBU%3D&reserved=0), National Center of Immunization and Respiratory Diseases (NCIRD) (contract number: 75D30120C08008; PI, Robert A. Weinstein, MD). A staff member of the CDC had a role in the study design, but not in the data collection and analysis, decision to publish, or preparation of the manuscript. The funders otherwise had no role in the study

## Objective

To compare patient-reported outcomes by employment type and SARS-CoV-2 status.

## Design

Data from the INSPIRE registry, which enrolled COVID-positive and COVID-negative adults between 12/7/2020–8/29/2022 was analyzed. Patient-reported outcomes were collected quarterly over 18 months.

## Setting

Participants were recruited across eight US sites.

## Participants

Employed INSPIRE participants who completed a short (3-month) and long-term (12–18 month) survey.

## Exposure

SARS-CoV-2 index status and employment type (essential healthcare worker [HCW], essential non-HCW, and non-essential worker ["general worker"]).

## Main outcomes and measures

PROMIS-29 (mental and physical health summary) and PROMIS Cognitive SF-CF 8a (cognitive function) scores were assessed at baseline, short-term (3-months), and long-term (12–18 months) timepoints using GEE modeling.

## Results

Of the 1,463 participants: 53.5% were essential workers (51.4% HCWs, 48.6% non-HCWs) and 46.5% were general workers. Most associations between outcomes and employment type became non-significant after adjusting for sociodemographics, comorbidities, COVID-19 vaccination, and SARS-CoV-2 variant period. However, among COVID-negative participants, essential HCWs had higher cognitive scores at baseline (β: 3.91, 95% CI [1.32, 6.50]), short term: (β: 3.49, 95% CI: [0.80, 6.18]) and long-term: (β: 3.72, 95% CI: [0.98, 6.46]) compared to general workers. Among COVID-positive participants, essential non-HCWs had significantly worse long-term physical health summary scores (β:-1.22, 95% CI: [−2.35, −0.09]) compared to general workers.

## Conclusions and relevance

Differences in outcomes by worker status were largely explained by baseline characteristics. However, compared to general workers, essential HCW status had higher cognitive function in the absence of SARS-CoV-2 infection at all timepoints,

design, data collection and analysis, decision to publish, or preparation of the manuscript.

**Competing interests:** I have read the journal's policy and the authors of this manuscript have the following competing interests: Lauren E. Wisk, PhD, reported receiving grants from the CDC during the conduct of the study and from the National Institutes of Health outside the submitted work. Ahamed H. Idris, MD, reported receiving grants from the CDC during the conduct of the study. Joann Elmore, MD, MPH, reported receiving grants from the CDC during the conduct of the study and serving as editor-in-chief of adult primary care for UpToDate and as director of the National Clinician Scholars Program at the University of California, Los Angeles. Kelli N. O'Laughlin, MD, MPH, reported a grant PROCOVAXED funded by NIAID R01AI166967-01 (PI: Rodriguez). Michael Gottlieb, MD MG, reports grant funding from the Rush Center for Emerging Infectious Diseases Research Grant, Biomedical Advanced Research and Development Authority Research Grant, Emergency Medicine Foundation/Council of Residency Directors in Emergency Medicine Education Research Grant, Emergency Medicine: Reviews and Perspectives Medical Education Research Grant, University of Ottawa Department of Medicine Education Grant; and Society of Directors of Research in Medical Education Grant. Nicole L. Gentile, MD, PhD, reported receiving grants from the CDC during the conduct of the study and being the primary care medical director of the Long COVID Clinic at the University of Washington. Robert M. Rodriguez, MD reported a grant PROCOVAXED funded by NIAID R01AI166967-01 (PI: Rodriguez). Ryan Huebinger, MD Reported receiving grants from the CDC during the conduct of the study. Mandy Hill, DrPH, MPH, reported an Investigator Award from Merck, MISP 100099, PI: Hill. Erica S. Spatz, MD, MHS, reported receiving grant funding from the CDC; US Food and Drug Administration to support projects within the Yale University-Mayo Clinic Center of Excellence in Regulatory Science and Innovation; National Heart, Lung, and Blood Institute; and Patient Centered Outcomes Research Institute. There are no patents, products in development or marketed products associated with this research to declare. This does not alter our adherence to PLOS ONE policies on sharing data and materials.

while essential non-HCWs were most vulnerable to poor recovery in long-term physical health following SARS-CoV-2 infection. Preparation efforts for future pandemics may consider enhanced protection and post-infection resources for frontline workers.

## Introduction

The COVID-19 pandemic has profoundly impacted millions of lives worldwide. By June 2023, the global count of confirmed COVID-19 cases exceeded 767 million, with over 6.9 million fatalities. Healthcare workers have borne the brunt of the pandemic's effects, with the World Health Organization (WHO) reporting that over 14% of COVID-19 cases involve this group [1]. In the United States (U.S.) frontline workers played a pivotal role in sustaining the nation's operations throughout the COVID-19 pandemic, which was officially declared an international public health crisis by the World Health Organization on January 30, 2020 [2]. The diverse group of essential workers encompassed not only healthcare workers (HCWs) but a wide range of non-HCWs, including public safety officials; maintenance personnel; and grocery, agricultural, food production, and delivery workers, comprising approximately one-third of the U.S. workforce [3]. Frontline workers worldwide risked their personal and household's health and well-being, with ongoing exposure to severe acute respiratory syndrome coronavirus 2 (SARS-CoV-2) and shortages of personal protective equipment (PPE) and staff [4,5]. Between January 1, 2020 and October 12, 2021, 440,044 HCWs in the US contracted SARS-CoV-2 among whom 1,469 died from infection [6].

Among the 18.6 million essential healthcare workers in the US, 40% identify as racial minorities, with Black and Hispanic workers more likely to serve in inpatient hospitals, assisted living, and home healthcare—settings with the highest infection risk. Moreover, essential non-HCWs also experienced insufficient protection from SARS-CoV-2, with more frequent PPE shortages and reports from workers of being unable to take off work to avoid exposure (e.g., during warehouse outbreaks) [7–11]. A nationwide survey of healthcare workers revealed psychological distress following SARS-CoV-2 infection with higher levels of anxiety, depressive symptoms, and burnout among those who contracted SARS-CoV-2 compared to those who did not [12,13]. Together, this evidence begs the question whether occupation, compounded by ongoing exposure to the virus and early return-to-work policies, have contributed to long-term physical and mental health disparities among all types of frontline workers. Little is known about differences in short- and long-term health effects of SARS-CoV-2 infections by essential worker status and type of employment [14–16].

To address this gap, we utilized data from the Innovative Support for Patients with SARS-CoV-2 Infections Registry (INSPIRE) study to compare baseline, short (3-month) and long-term (12–18 month) health outcomes of essential workers (HCWs and non-HCWs) and non-essential workers during the COVID-19 pandemic.

## Methods

### Study design and data

INSPIRE is a multicenter prospective longitudinal registry study that enrolled U.S. adults with COVID-like symptoms across 8 research sites [16]. The enrollment period for the study varied by institution, with all sites concluding enrollment on August 29, 2022 (**Table 1**).

The study prospectively enrolled symptomatic adults testing either positive (COVID-positive) or negative (COVID-negative) for SARS-CoV-2 in a 3:1 ratio and collected electronic surveys every three months over 18-months. Inclusion criteria included age ≥ 18 years, fluency in English or Spanish, self-reported symptoms suggestive of acute SARS-CoV-2 infection at time of testing, and testing for SARS-CoV-2 with an FDA-approved/authorized molecular or antigen-based assay within the preceding 42 days. Exclusion criteria included inability to provide consent, imprisonment, unconfirmed SARS-CoV-2 test results, having a previous SARS-CoV-2 infection >42 days before enrollment, and lacking access to an internet-connected device for electronic survey completion. A total of 8,950 participants enrolled in the study between 12/7/2020 and 8/29/2022, of which 6,044 were eligible for follow-up based on the three sets of eligibility criteria above. This analysis included participants who reported being employed at baseline and completed the 3-month and at least one long-term (12, 15, or 18 month) survey. Informed consent was obtained electronically from all participants and stored on an electronic health management system, Hugo Health. Study coordinators spoke to participants over the phone and explained the consent form, study participation activities, and answered any questions participants had. The study was approved by the Institutional Review Boards of each participating site including Rush University (IRB#20030902-IRB01), Yale University (IRB#2000027976), the University of Washington (IRB#STUDY00009920), Thomas Jefferson University (IRB##20P.1150), the University of Texas Southwestern (IRB#STU-202–1352), the University of Texas, Houston (IRB#HSC-MS-20–0981), the University of California, San Francisco (IRB#20–32222) and the University of California, Los Angeles (IRB#20–001683).

### Exposures

Employment status and type were determined via baseline survey responses. The survey asked, "Were you employed before the coronavirus outbreak? (Yes/No)". If yes, they were asked "Did you have a change in your job status since COVID?" (No change; Reduced work hours; Permanently lost job; Temporarily lost job/Furloughed; Increased work hours)." Those not employed before the pandemic or reporting subsequent job loss were excluded from analysis. Baseline employment type was determined by the following yes or no questions: "Do you work in a healthcare setting such as a hospital, clinic, or nursing/rehabilitation care facility?" and if no, "Are you a non-health essential worker who was asked to work outside the home throughout the epidemic?" Participants who worked in healthcare settings were assigned to the essential HCW group, while participants who worked in non-healthcare settings were assigned to the essential non-HCW

**Table 1. Institutions start date of enrollment in the INSPIRE Study.**

| Institution | Start Date |
| --- | --- |
| Rush University | November 17, 2020 |
| University of Washington | December 11, 2020 |
| Yale University | December 26, 2020 |
| University of California, Los Angeles | February 1, 2021 |
| Thomas Jefferson University | February 8, 2021 |
| University of California, San Francisco (UCSF) | February 24, 2021 |
| University of Texas Southwestern (UTSW) | April 21, 2021 |
| University of Texas Health Science Center at Houston (UTH) | May 5, 2021 |

group. Participants who responded "no" to both essential work questions were assigned to the non-essential worker ("general worker") group. COVID status (COVID-positive or COVID-negative) was established based on results of index SARS-CoV-2 test, regardless of new infections reported during the follow-up period.

### Descriptive variables

Participant characteristics were established via baseline and 3-month survey responses. Sociodemographics, location of index SARS-CoV-2 test, and history of COVID-19 vaccination were collected at baseline; comorbidities were collected at 3-months [17]. SARS-CoV-2 variant period was approximated based on the predominant variant at index test date, regardless of test result [18].

### Patient-reported outcomes

PROMIS®-29 was used to measure physical and mental well-being and PROMIS® SF-CF 8a surveys were used to measure cognitive function [19,20]. The PROMIS®-29 instrument is divided into 7 subscales: physical function, fatigue, pain interference and intensity, depressive symptoms, anxiety, ability to participate in social roles and activities, and sleep disturbance. The scores calculated from the surveys were adjusted to a standardized scale called T-scores. The scale has a mean of 50, with a higher T-score representing more of the measured concept. For example, a higher score for anxiety indicates greater anxiety, while a higher score for physical function indicates better mobility. For evaluating cognition, a higher score indicates better function; this score is also adjusted to a standardized T-score with a mean of 50 [21]. The general guidelines for PROMIS Scores provide cut-off points to interpret minimally important changes (MICs) to T-scores. MICs in T scores typically range between 2 and 6 points and we considered MIC > 2 to be clinically significant [22]. We reported the observed T-scores (mean with standard deviation [SD]), and the prevalence of poor-to-very-poor physical health (summary score < 42), mental health (summary score < 40) and cognitive function (score < 40) [23].

For this analysis, we report results from PROMIS-29® and PROMIS® Cognitive SF-8 in 3 domains of well-being: physical health summary score, mental health summary score, and cognitive function. The physical and mental health summary scores are the sum of domain scores weighted by corresponding factor loadings from the published confirmatory factor analysis (CFA) study [24]. The factor loadings were either negative or weak (<.01) for the adverse domains (e.g., fatigue, pain) and positive for desired domains, so higher physical and mental health summary scores are considered better outcomes.

### Statistical analysis

We compared participant characteristics by employment type and SARS-CoV-2 status using chi-square tests. To model the association between employment type and outcomes and to account for repeated measures of the outcomes across time, we used generalized estimating equations (GEE), adjusting for (1) sociodemographic variables, SARS-CoV-2 variant period at index test, and comorbidities, and (2) interactions between employment and timepoint, employment and index test status, timepoint and index test status. To account for the non-linear trajectory of the outcomes, timepoint was modeled as a categorical variable. We also ran unadjusted GEE models without the sociodemographic variables, SARS-CoV-2 variant period at index test, and comorbidities. Based on the unadjusted and adjusted GEE models, we determined the marginal effects of physical, mental, and cognitive health for essential HCWs and essential non-HCWs using PROMIS-29 and PROMIS Cognitive SF8 outcomes compared to the reference group of general workers at baseline and short- and long-term follow-up stratified by index SARS-CoV-2 status. Since this is an exploratory study, we did not adjust for multiple comparisons [25]. Statistical tests (including F-tests and Wald tests for the coefficients) were two-sided with $\alpha = 0.05$; chi-square tests were one-sided with $\alpha = 0.05$. Statistical analyses were performed using SAS 9.4 (SAS Institute Inc., Cary, NC) and R v4.2.2 (R Foundation for Statistical Computing). Additional information about our GEE methods is available in **Appendix 2** in **S1 Appendix**.

## Results

### Participant characteristics

Among 6,044 study participants, 2,130 did not meet employment criteria and 2,451 did not meet survey completion criteria, leaving 1,463 participants in the final analytic cohort (**Fig 1**). In order to maintain strength of the sample size COVID-positive participants with reinfection (n = 311) or COVID-negative participants with new infection (n = 155) during the follow-up period were still included in the analysis. Approximately half of participants (53.5%) were essential workers, of which 51.4% were HCWs and 48.6% were non-HCWs. Three-quarters of participants tested positive for SARS-CoV-2 at baseline (77.4%) and most participants enrolled during the delta variant period (57.6%), while 23.4% enrolled before (pre-delta) and 19% enrolled during the omicron variant period. Overall, participants were young (39.8% aged 18–34 years), predominately female (65.3%), White (70.1%), and non-Hispanic (84.1%), and most had at least 4 years of college education (72.2%) and private health insurance (83.3%). Highly reported comorbidities included obesity (26.8%), hypertension (13.1%), asthma (12.1%), diabetes (4.4%) and smoking (3.8%) (**Table 2**).

Stratification by exposure groups revealed significant differences in baseline characteristics. Across employment groups, there were significant differences in sociodemongraphic characteristics as well as diabetes, obesity and smoking . When further stratifying by SARS-CoV-2 status (**Table 3**), the COVID-negative group had a slightly higher proportion of essential worker HCWs than the COVID-positive group (33.0% vs. 25.9%, respectively) and the COVID-positive group had a higher proportion of essential non-HCWs than the COVID-negative group (26.8% vs. 23.0%, respectively). Age, gender, education, income, health insurance, and COVID-19 vaccination differed within both COVID groups, while race and COVID-19 variant differed within the COVID-positive group only. There were no differences in ethnicity and most differences in comorbidities were not significant.

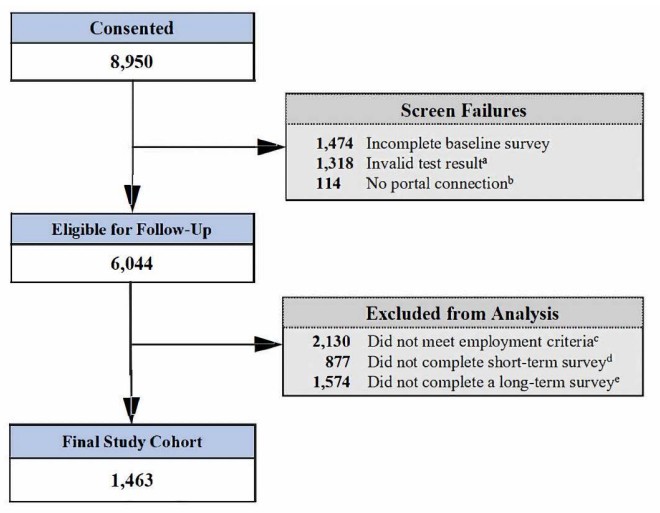

Footnote:
ᵃ Invalid test result = no proof of SARS-CoV-2 test or had positive test >42 days ago.
ᵇ Portal connection was required for follow-up eligibility through 3/21/22.
ᶜ Reported unemployment or temporary or permanent job loss on baseline survey.
ᵈ Incomplete 3-month survey due to withdrawal, death, or loss to follow-up.
ᵉ Incomplete 12, 15, and 18-month surveys due to withdrawal, death, end-of-study censoring, or loss to follow-up.

**Fig 1. Participant flowchart.**

**Table 2. Participant characteristics by employment category.**

| Participant Characteristic | Employment Category | | | | |
|---|---|---|---|---|---|
| | Total<br>N | Essential<br>HCW<br>N (%) | Essential<br>Non-HCW<br>N (%) | General<br>Worker<br>N (%) | P-value |
| **Participants** | 1,463 | 402 (27.5) | 380 (26.0) | 681 (46.5) | |
| **Age** | | | | | |
| 18–34 | 582 (39.8) | 188 (47.2) | 131 (34.7) | 263 (38.8) | <0.001* |
| 35–49 | 513 (35.1) | 136 (34.2) | 126 (33.3) | 251 (37.1) | |
| 50–64 | 292 (20.0) | 65 (16.3) | 105 (27.8) | 122 (18.0) | |
| 65+ | 66 (4.5) | 9 (2.3) | 16 (4.2) | 41 (6.1) | |
| Missing | 10 (0.7) | 4 (1.0) | 2 (0.5) | 4 (0.6) | |
| **Gender** | | | | | |
| Female | 956 (65.3) | 307 (79.5) | 232 (62.5) | 417 (62.8) | <0.001* |
| Male | 450 (30.8) | 74 (19.2) | 135 (36.4) | 241 (36.3) | |
| Transgender/Non-binary/Other | 15 (1.0) | 5 (1.3) | 4 (1.1) | 6 (0.9) | |
| Missing | 42 (2.9) | 16 (4.0) | 9 (2.4) | 17 (2.5) | |
| **Ethnicity** | | | | | |
| Non-Hispanic | 1230 (84.1) | 335 (85.7) | 305 (81.3) | 590 (87.9) | 0.014* |
| Hispanic/Latino | 207 (14.1) | 56 (14.3) | 70 (18.7) | 81 (12.1) | |
| Missing | 26 (1.8) | 11 (2.7) | 5 (1.3) | 10 (1.5) | |
| **Race** | | | | | |
| White | 1025 (70.1) | 261 (67.6) | 278 (74.9) | 486 (73.1) | <0.001* |
| Black | 111 (7.6) | 38 (9.8) | 33 (8.9) | 40 (6.0) | |
| Asian | 169 (11.6) | 57 (14.8) | 23 (6.2) | 89 (13.4) | |
| Other/Multiple | 117 (8.0) | 30 (7.8) | 37 (10.0) | 50 (7.5) | |
| Missing | 41 (2.8) | 16 (4.0) | 9 (2.4) | 16 (2.3) | |
| **Education** | | | | | |
| Less than high school | 10 (0.7) | 1 (0.3) | 5 (1.3) | 4 (0.6) | <0.001* |
| High school graduate | 71 (4.9) | 20 (5.1) | 25 (6.7) | 26 (3.9) | |
| Some College | 183 (12.5) | 42 (10.7) | 72 (19.4) | 69 (10.3) | |
| 2-year degree | 113 (7.7) | 38 (9.7) | 37 (10.0) | 38 (5.7) | |
| 4-year degree | 487 (33.3) | 124 (31.6) | 107 (28.8) | 256 (38.2) | |
| More than 4 years | 570 (39.0) | 168 (42.7) | 125 (33.7) | 277 (41.3) | |
| Missing | 29 (2.0) | 9 (2.2) | 9 (2.4) | 11 (1.6) | |
| **Household Income** | | | | | |
| <$10,000 | 40 (2.7) | 12 (3.0) | 11 (2.9) | 17 (2.5) | <0.001* |
| $10,000–34,999 | 118 (8.1) | 29 (7.2) | 43 (11.3) | 46 (6.8) | |
| $35,000–49,999 | 133 (9.1) | 45 (11.2) | 52 (13.7) | 36 (5.3) | |
| $50,000–74,999 | 219 (15.0) | 55 (13.7) | 68 (17.9) | 96 (14.1) | |
| ≥$75,000+ | 878 (60.0) | 240 (59.7) | 186 (48.9) | 452 (66.4) | |
| Prefer not to answer | 75 (5.1) | 21 (5.2) | 20 (5.3) | 34 (5.0) | |
| **Health Insurance** | | | | | |
| Private only | 1221 (83.5) | 360 (89.6) | 298 (78.4) | 563 (82.7) | <0.001* |
| Public only | 162 (11.1) | 30 (7.5) | 59 (15.5) | 73 (10.7) | |
| Private and public | 43 (2.9) | 8 (2.0) | 10 (2.6) | 25 (3.7) | |
| None | 37 (2.5) | 4 (1.0) | 13 (3.4) | 20 (2.9) | |

*(Continued)*

**Table 2.** (Continued)

| Participant Characteristic | Employment Category | | | | |
|---|---|---|---|---|---|
| | Total N | Essential HCW N (%) | Essential Non-HCW N (%) | General Worker N (%) | P-value |
| **Comorbidities** | | | | | |
| Asthma | 177 (12.1) | 53 (13.6) | 49 (13.1) | 75 (11.2) | 0.45 |
| Kidney disease | 16 (1.1) | 2 (0.5) | 7 (1.9) | 7 (1.0) | 0.19 |
| Emphysema (COPD) | 4 (0.3) | 0 (0.0) | 3 (0.8) | 1 (0.1) | 0.07 |
| Heart conditions | 22 (1.5) | 6 (1.5) | 7 (1.9) | 9 (1.3) | 0.8 |
| Diabetes | 65 (4.4) | 18 (4.6) | 29 (7.8) | 18 (2.7) | <0.001* |
| Hypertension | 192 (13.1) | 47 (12.0) | 61 (16.4) | 84 (12.5) | 0.14 |
| Liver disease | 13 (0.9) | 4 (1.0) | 5 (1.3) | 4 (0.6) | 0.46 |
| Obesity | 392 (26.8) | 96 (24.6) | 129 (34.6) | 167 (24.9) | 0.001* |
| Smoking | 55 (3.8) | 14 (3.6) | 24 (6.4) | 17 (2.5) | 0.007* |
| Missing | 28 (1.9) | 11 (2.7) | 7 (1.8) | 10 (1.5) | |
| **COVID-19 Vaccination Status** | | | | | |
| Vaccinated | 871 (59.5) | 274 (74.9) | 195 (59.6) | 402 (66.0) | <0.001* |
| Unvaccinated | 431 (29.5) | 92 (25.1) | 132 (40.4) | 207 (34.0) | |
| Missing | 161 (11.0) | 36 (9.0) | 53 (13.9) | 72 (10.6) | |
| **COVID-19 Variant at Index Test** | | | | | |
| Pre-Delta | 342 (23.4) | 87 (21.6) | 107 (28.2) | 148 (21.7) | <0.001* |
| Delta | 843 (57.6) | 209 (52.0) | 205 (53.9) | 429 (63.0) | |
| Omicron | 278 (19.0) | 106 (26.4) | 68 (17.9) | 104 (15.3) | |
| **Index COVID-19 Status** | | | | | |
| Positive | 1133 (77.4) | 293 (72.9) | 304 (80.0) | 536 (78.7) | 0.033* |
| Negative | 330 (22.6) | 109 (27.1) | 76 (20.0) | 145 (21.3) | |

*P-value < 0.05, indicating that statistically significant difference in each characteristic using chi-square tests of association. Index COVID-19 vaccination status was obtained from a combination of linked electronic health record data and survey responses and indicates at least one dose prior to the index SARS-CoV-2 test. Table excludes participants who did not meet survey completion criteria (N = 2,451) or employment criteria (N = 2,130) and chi-square tests excluded participants with any responses of missing or prefer not to answer.

## Outcomes

Within COVID groups, most differences in mean scores were non-significant between employment groups, except for physical health scores (all timepoints) and short-term mental health among COVID-positive participants, and baseline and short-term cognitive function among COVID-negative participants. Similarly, within COVID groups, few differences in prevalence of poor-to-very-poor outcomes were observed between employment groups, except for short-term physical health in COVID-positive participants and short-term cognitive function in COVID-negative participants. Among COVID-positive participants, essential non-HCWs reported the highest prevalence of poor-to-very-poor short-term physical health outcomes compared to other groups (18.1% vs. 9.5–15%). Among COVID-negative participants, essential HCWs reported the lowest prevalence of poor-to-very-poor short-term cognitive function compared to other groups (23.9% vs. 37.9–38.2%) (Table 4).

In unadjusted analyses of COVID-positive participants, essential non-HCWs had worse physical health scores at all timepoints (baseline: β −1.78, 95% CI [−2.75, −0.80], short-term: β −1.67, 95% CI [−2.73, −0.61], long-term: β −1.78 95% CI [−2.85, −0.71]) compared to general workers (Fig 2). With respect to mental health, essential HCWs had worse short-term scores (β −1.27, 95% CI [−2.44, −0.09]) and essential non-HCWs had worse long-term scores (β-1.25

**Table 3. Participant characteristics by employment category stratified by SARS-CoV-2 status.**

| | COVID-POSITIVE | | | | | COVID-NEGATIVE | | | | |
|---|---|---|---|---|---|---|---|---|---|---|
| | Total N | Essential HCW N (%) | Essential Non-HCW N (%) | General worker N (%) | P-value | Total N | Essential HCW N (%) | Essential Non-HCW N (%) | General worker N (%) | P-value |
| **Participants Characteristics** | 1,133 | 293 (25.9) | 304 (26.8) | 536 (47.3) | | 330 | 109 (33.0) | 76 (23.0) | 145 (43.9) | |
| **Age** | | | | | | | | | | |
| 18–34 | 438 (38.7) | 132 (45.7) | 103 (34.0) | 203 (38.1) | 0.005* | 144 (43.6) | 56 (51.4) | 28 (37.3) | 60 (41.7) | 0.01* |
| 35–49 | 400 (35.3) | 100 (34.6) | 104 (34.3) | 196 (36.8) | | 113 (34.2) | 36 (33.0) | 22 (29.3) | 55 (38.2) | |
| 50–64 | 239 (21.1) | 50 (17.3) | 84 (27.7) | 105 (19.7) | | 53 (16.1) | 15 (13.8) | 21 (28.0) | 17 (11.8) | |
| 65+ | 48 (4.2) | 7 (2.4) | 12 (4.0) | 29 (5.4) | | 18 (5.5) | 2 (1.8) | 4 (5.3) | 12 (8.3) | |
| Missing | 8 (0.7) | 4 (1.4) | 1 (0.3) | 3 (0.6) | | 2 (0.6) | 0 (0.0) | 1 (1.3) | 1 (0.7) | |
| **Gender** | | | | | | | | | | |
| Female | 713 (62.9) | 217 (77.5) | 177 (59.6) | 319 (61.2) | <0.001* | 243 (73.6) | 90 (84.9) | 55 (74.3) | 98 (68.5) | 0.029* |
| Male | 375 (33.1) | 59 (21.1) | 116 (39.1) | 200 (38.4) | | 75 (22.7) | 15 (14.2) | 19 (25.7) | 41 (28.7) | |
| Transgender/ Nonbinary/Other | 10 (0.9) | 4 (1.4) | 4 (1.3) | 2 (0.4) | | 5 (1.5) | 1 (0.9) | 0 (0.0) | 4 (2.8) | |
| Missing | 35 (3.1) | 13 (4.4) | 7 (2.3) | 15 (2.8) | | 7 (2.1) | 3 (2.8) | 2 (2.6) | 2 (1.4) | |
| **Ethnicity** | | | | | | | | | | |
| Non-Hispanic | 959 (84.6) | 246 (86.3) | 247 (82.3) | 466 (88.4) | 0.05 | 271 (82.1) | 89 (84.0) | 58 (77.3) | 124 (86.1) | 0.25 |
| Hispanic/Latino | 153 (13.5) | 39 (13.7) | 53 (17.7) | 61 (11.6) | | 54 (16.4) | 17 (16.0) | 17 (22.7) | 20 (13.9) | |
| Missing | 21 (1.9) | 8 (2.7) | 4 (1.3) | 9 (1.7) | | 5 (1.5) | 3 (2.8) | 1 (1.3) | 1 (0.7) | |
| **Race** | | | | | | | | | | |
| White | 808 (71.3) | 192 (68.3) | 224 (74.9) | 392 (74.7) | 0.003* | 217 (65.8) | 69 (65.7) | 54 (75.0) | 94 (67.1) | 0.19 |
| Black | 81 (7.1) | 28 (10.0) | 24 (8.0) | 29 (5.5) | | 30 (9.1) | 10 (9.5) | 9 (12.5) | 11 (7.9) | |
| Asian | 124 (10.9) | 42 (14.9) | 19 (6.4) | 63 (12.0) | | 45 (13.6) | 15 (14.3) | 4 (5.6) | 26 (18.6) | |
| Other/Multiple | 92 (8.1) | 19 (6.8) | 32 (10.7) | 41 (7.8) | | 25 (7.6) | 11 (10.5) | 5 (6.9) | 9 (6.4) | |
| Missing | 28 (2.5) | 12 (4.1) | 5 (1.6) | 11 (2.1) | | 13 (3.9) | 4 (3.7) | 4 (5.3) | 5 (3.4) | |
| **Education** | | | | | | | | | | |
| Less than High school | 7 (0.6) | 1 (0.3) | 4 (1.3) | 2 (0.4) | <0.001* | 3 (0.9) | 0 (0.0) | 1 (1.4) | 2 (1.4) | 0.03* |
| High school graduate | 57 (5.0) | 16 (5.6) | 18 (6.1) | 23 (4.3) | | 14 (4.2) | 4 (3.7) | 7 (9.5) | 3 (2.1) | |
| Some College | 143 (12.6) | 29 (10.1) | 57 (19.2) | 57 (10.8) | | 40 (12.1) | 13 (12.1) | 15 (20.3) | 12 (8.5) | |
| 2-year degree | 92 (8.1) | 30 (10.5) | 30 (10.1) | 32 (6.0) | | 21 (6.4) | 8 (7.5) | 7 (9.5) | 6 (4.3) | |
| 4-year degree | 393 (34.7) | 91 (31.8) | 93 (31.3) | 209 (39.5) | | 94 (28.5) | 33 (30.8) | 14 (18.9) | 47 (33.3) | |
| More than 4 years | 420 (37.1) | 119 (41.6) | 95 (32.0) | 206 (38.9) | | 150 (45.5) | 49 (45.8) | 30 (40.5) | 71 (50.4) | |
| Missing | 21 (1.9) | 7 (2.4) | 7 (2.3) | 7 (1.3) | | 8 (2.4) | 2 (1.8) | 2 (2.6) | 4 (2.8) | |
| **Household Income** | | | | | | | | | | |
| <$10,000 | 26 (2.3) | 9 (3.1) | 6 (2.0) | 11 (2.1) | <0.001* | 14 (4.2) | 3 (2.8) | 5 (6.6) | 6 (4.1) | 0.003* |
| $10,000–34,999 | 94 (8.3) | 22 (7.5) | 34 (11.2) | 38 (7.1) | | 24 (7.3) | 7 (6.4) | 9 (11.8) | 8 (5.5) | |
| $35,000–49,999 | 93 (8.2) | 27 (9.2) | 39 (12.8) | 27 (5.0) | | 40 (12.1) | 18 (16.5) | 13 (17.1) | 9 (6.2) | |
| $50,000–74,999 | 167 (14.7) | 43 (14.7) | 55 (18.1) | 69 (12.9) | | 52 (15.8) | 12 (11.0) | 13 (17.1) | 27 (18.6) | |
| ≥$75,000 | 700 (61.8) | 179 (61.1) | 159 (52.3) | 362 (67.5) | | 178 (53.9) | 61 (56.0) | 27 (35.5) | 90 (62.1) | |
| Prefer not to answer | 53 (4.7) | 13 (4.4) | 11 (3.6) | 29 (5.4) | | 22 (6.7) | 8 (7.3) | 9 (11.8) | 5 (3.4) | |

*(Continued)*

**Table 3.** (Continued)

| | COVID-POSITIVE | | | | | COVID-NEGATIVE | | | | |
|---|---|---|---|---|---|---|---|---|---|---|
| | Total N | Essential HCW N (%) | Essential Non-HCW N (%) | General worker N (%) | P-value | Total N | Essential HCW N (%) | Essential Non-HCW N (%) | General worker N (%) | P-value |
| **Health Insurance** | | | | | | | | | | |
| Private only | 944 (83.3) | 258 (88.1) | 243 (79.9) | 443 (82.6) | 0.027* | 277 (83.9) | 102 (93.6) | 55 (72.4) | 120 (82.8) | 0.011* |
| Public only | 122 (10.8) | 25 (8.5) | 44 (14.5) | 53 (9.9) | | 40 (12.1) | 5 (4.6) | 15 (19.7) | 20 (13.8) | |
| Private and public | 34 (3.0) | 6 (2.0) | 6 (2.0) | 22 (4.1) | | 9 (2.7) | 2 (1.8) | 4 (5.3) | 3 (2.1) | |
| None | 33 (2.9) | 4 (1.4) | 11 (3.6) | 18 (3.4) | | 4 (1.2) | 0 (0.0) | 2 (2.6) | 2 (1.4) | |
| **Comorbidities** | | | | | | | | | | |
| Asthma | 129 (11.4) | 38 (13.5) | 33 (11.1) | 58 (11.0) | 0.54 | 48 (14.5) | 15 (13.8) | 16 (21.1) | 17 (11.9) | 0.18 |
| Kidney disease | 11 (1.0) | 1 (0.4) | 6 (2.0) | 4 (0.8) | 0.1 | 5 (1.5) | 1 (0.9) | 1 (1.3) | 3 (2.1) | 0.74 |
| Emphysema (COPD) | 3 (0.3) | 0 (0.0) | 2 (0.7) | 1 (0.2) | 0.26 | 1 (0.3) | 0 (0.0) | 1 (1.3) | 0 (0.0) | 0.19 |
| Heart conditions | 15 (1.3) | 4 (1.4) | 5 (1.7) | 6 (1.1) | 0.8 | 7 (2.1) | 2 (1.8) | 2 (2.6) | 3 (2.1) | 0.93 |
| Diabetes | 46 (4.1) | 12 (4.3) | 20 (6.7) | 14 (2.7) | 0.019* | 19 (5.8) | 6 (5.5) | 9 (11.8) | 4 (2.8) | 0.024* |
| Hypertension | 140 (12.4) | 32 (11.3) | 48 (16.2) | 60 (11.4) | 0.1 | 52 (15.8) | 15 (13.8) | 13 (17.1) | 24 (16.8) | 0.76 |
| Liver disease | 9 (0.8) | 4 (1.4) | 3 (1.0) | 2 (0.4) | 0.26 | 4 (1.2) | 0 (0.0) | 2 (2.6) | 2 (1.4) | 0.27 |
| Obesity | 297 (26.2) | 69 (24.5) | 101 (34.0) | 127 (24.1) | 0.005* | 95 (28.8) | 27 (24.8) | 28 (36.8) | 40 (28.0) | 0.19 |
| Smoking | 41 (3.6) | 11 (3.9) | 15 (5.1) | 15 (2.8) | 0.27 | 14 (4.2) | 3 (2.8) | 9 (11.8) | 2 (1.4) | <0.001* |
| Missing | 26 (2.3) | 11 (3.8) | 7 (2.3) | 8 (1.5) | | 2 (0.6) | 0 (0.0) | 0 (0.0) | 2 (1.4) | |
| **COVID-19 Variant at index test** | | | | | | | | | | |
| Pre-Delta | 242 (21.4) | 54 (18.4) | 83 (27.3) | 105 (19.6) | <0.001* | 100 (30.3) | 33 (30.3) | 24 (31.6) | 43 (29.7) | 0.16 |
| Delta | 679 (59.9) | 163 (55.6) | 166 (54.6) | 350 (65.3) | | 164 (49.7) | 46 (42.2) | 39 (51.3) | 79 (54.5) | |
| Omicron | 212 (18.7) | 76 (25.9) | 55 (18.1) | 81 (15.1) | | 66 (20.0) | 30 (27.5) | 13 (17.1) | 23 (15.9) | |
| **COVID-19 Vaccination Status** | | | | | | | | | | |
| Vaccinated | 650 (57.4) | 192 (72.5) | 157 (59.2) | 301 (62.8) | 0.004* | 221 (67.0) | 82 (81.2) | 38 (61.3) | 101 (77.7) | 0.012* |
| Unvaccinated | 359 (31.7) | 73 (27.5) | 108 (40.8) | 178 (37.2) | | 72 (21.8) | 19 (18.8) | 24 (38.7) | 29 (22.3) | |
| Missing | 124 (10.9) | 28 (9.6) | 39 (12.8) | 57 (10.6) | | 37 (11.2) | 8 (7.3) | 14 (18.4) | 15 (10.3) | |

*P-value < 0.05, indicating that statistically significant difference in each characteristic using chi-square tests of association. Index COVID-19 vaccination status was obtained from a combination of linked electronic health record data and survey responses and indicates at least one dose prior to the index SARS-CoV-2 test. Table excludes participants who did not meet survey completion criteria (N = 2,451) or employment criteria (N = 2,130) and chi-square tests excluded participants with any responses of missing or prefer not to answer.

[−2.41, −0.10]) compared with general workers. There were no significant differences in physical health among essential HCWs or in cognitive function among any worker group compared to general workers. After adjusting for confounders, only the difference in long-term physical health remained significant between essential non-HCWs and general workers (β −1.22, 95% CI [−2.35, −0.09]), although the clinical significance of differences was modest (MIC score < 2 points).

In both unadjusted and adjusted analyses of COVID-negative participants, there were no significant differences in physical or mental health scores between essential worker groups compared to general workers. However, unadjusted analyses showed essential HCWs reported significantly better cognitive scores at baseline, short-term, and long-term follow-up compared with general workers. This association was even more pronounced after adjusting for confounders,

**Table 4. Summary statistics of T-Scores at baseline, 3-month and 12−18 months by employment category and SARS-CoV-2 status.**

| Outcome | Statistics | Total N = 1,463 | COVID-POSITIVE N = 1,133 | | | | COVID-NEGATIVE N = 330 | | | |
|---|---|---|---|---|---|---|---|---|---|---|
| | | | Essential HCW N = 293 | Essential Non-HCW N = 304 | General Worker N = 536 | P-value | Essential HCW N = 109 | Essential Non-HCW N = 76 | General Worker N = 145 | P-value |
| **Physical Health** | | | | | | | | | | |
| 0 m | Mean (SD) | 46.7 (10.1) | 48.1 (9.4) | 46.1 (10.2) | 47.9 (9.9) | 0.015* | 50.3 (8.5) | 48.6 (8.5) | 48.8 (8.4) | 0.3 |
| | % score<42 | 35.7 | 30.7 | 38.2 | 30.6 | 0.06 | 22 | 26.3 | 22.8 | 0.77 |
| 3 m | Mean (SD) | 51.4 (8.8) | 53.0 (7.5) | 52.3 (8.1) | 54.1 (6.8) | 0.002** | 52.2 (8.0) | 51.2 (8.6) | 50.3 (9.0) | 0.23 |
| | % score<42 | 20 | 15 | 18.1 | 9.5 | 0.001** | 14.7 | 19.7 | 22.1 | 0.33 |
| 12–18 m | Mean (SD) | 51.3 (8.8) | 52.9 (7.7) | 52.1 (8.2) | 53.8 (7.0) | 0.008** | 51.5 (8.5) | 50.6 (9.1) | 50.9 (8.9) | 0.77 |
| | % score<42 | 19.9 | 13.3 | 16.8 | 10.3 | 0.024 | 20.2 | 23.7 | 20 | 0.79 |
| **Mental Health** | | | | | | | | | | |
| 0 m | Mean (SD) | 46.9 (8.7) | 48.0 (8.4) | 47.8 (8.3) | 48.2 (8.3) | 0.78 | 47.2 (7.8) | 46.2 (8.5) | 46.7 (7.4) | 0.69 |
| | % score<40 | 23.1 | 18.4 | 20.4 | 17 | 0.47 | 18.3 | 28.9 | 17.9 | 0.12 |
| 3 m | Mean (SD) | 50.6 (9.2) | 51.4 (8.7) | 51.7 (9.0) | 53.0 (8.2) | 0.019** | 49.9 (8.8) | 49.1 (10.2) | 48.6 (8.6) | 0.53 |
| | % score<40 | 15.4 | 13.7 | 11.5 | 8.8 | 0.08 | 16.5 | 19.7 | 18.6 | 0.84 |
| 12–18 m | Mean (SD) | 51.4 (9.4) | 52.8 (8.8) | 52.4 (8.6) | 53.7 (8.2) | 0.1 | 50.9 (9.0) | 49.2 (10.1) | 50.1 (8.6) | 0.46 |
| | % score<40 | 12.4 | 7.8 | 8.9 | 5.4 | 0.13 | 12.8 | 22.4 | 11 | 0.06 |
| **Cognitive Function** | | | | | | | | | | |
| 0 m | Mean (SD) | 45.9 (11.0) | 47.3 (11.2) | 47.2 (11.1) | 46.1 (10.8) | 0.22 | 48.0 (11.0) | 44.3 (10.6) | 44.6 (10.2) | 0.022* |
| | % score<40 | 35.6 | 31.1 | 31.3 | 35.4 | 0.31 | 25.7 | 38.2 | 37.9 | 0.08 |
| 3 m | Mean (SD) | 47.7 (11.2) | 49.1 (11.1) | 48.7 (11.2) | 49.3 (10.9) | 0.72 | 48.8 (10.8) | 46.8 (11.6) | 44.9 (10.6) | 0.017** |
| | % score<40 | 30 | 27.6 | 27.3 | 22.8 | 0.19 | 23.9 | 38.2 | 37.9 | 0.037* |
| 12–18 m | Mean (SD) | 48.5 (11.6) | 50.4 (11.5) | 49.4 (11.5) | 49.9 (11.2) | 0.55 | 48.8 (11.5) | 47.5 (11.8) | 46.3 (11.5) | 0.23 |
| | % score<40 | 29.5 | 24.6 | 26 | 23.3 | 0.68 | 32.1 | 36.8 | 33.8 | 0.8 |

[a]% score<42 for physical health summary score indicates the percent of participants who reported score<42; % score<40 for mental health summary score indicates the percent of participants who reported score<40; % score<40 for cognitive function score indicates the percent of participants who reported score<40. Chi-square tests were conducted for these variables in each COVID-group at each timepoint to determine association with the worker variable.

[b]The observed means with corresponding standard deviations of physical health summary, mental health summary, and cognitive function score are reported, and F-tests have been conducted to determine the difference in means among the worker groups in each COVID-group at each timepoint.

[c]*P-value<0.05 indicating statistically significant results from the corresponding tests.

with essential HCWs reporting better cognitive scores at baseline (β 3.91, 95% CI [1.32, 6.50]), short-term (β 3.49, 95% CI [0.80, 6.18]) and long-term (β 3.72, 95% CI [0.98, 6.46]) follow-up compared to general workers. These differences were considered clinically significant at all timepoints (MIC score>2 points).

## Discussion

In this prospective, multicenter longitudinal study, we found varied differences in three domains of patient-reported outcomes (mental health, physical health, cognitive function) when comparing two categories of essential workers – HCWs and non-HCWs – with the general worker population. While a comparison of observed health scores revealed that essential workers had better short- and long-term cognitive function in the absence of a SARS-CoV-2 infection and poorer short- and long-term physical health following a SARS-CoV-2 infection compared to general workers, further analysis revealed that findings often differed by essential worker type, SARS-CoV-2 infection status and timepoint, identifying distinct association with SARS-CoV-2 infections and the COVID-19 pandemic on short and long-term well-being.

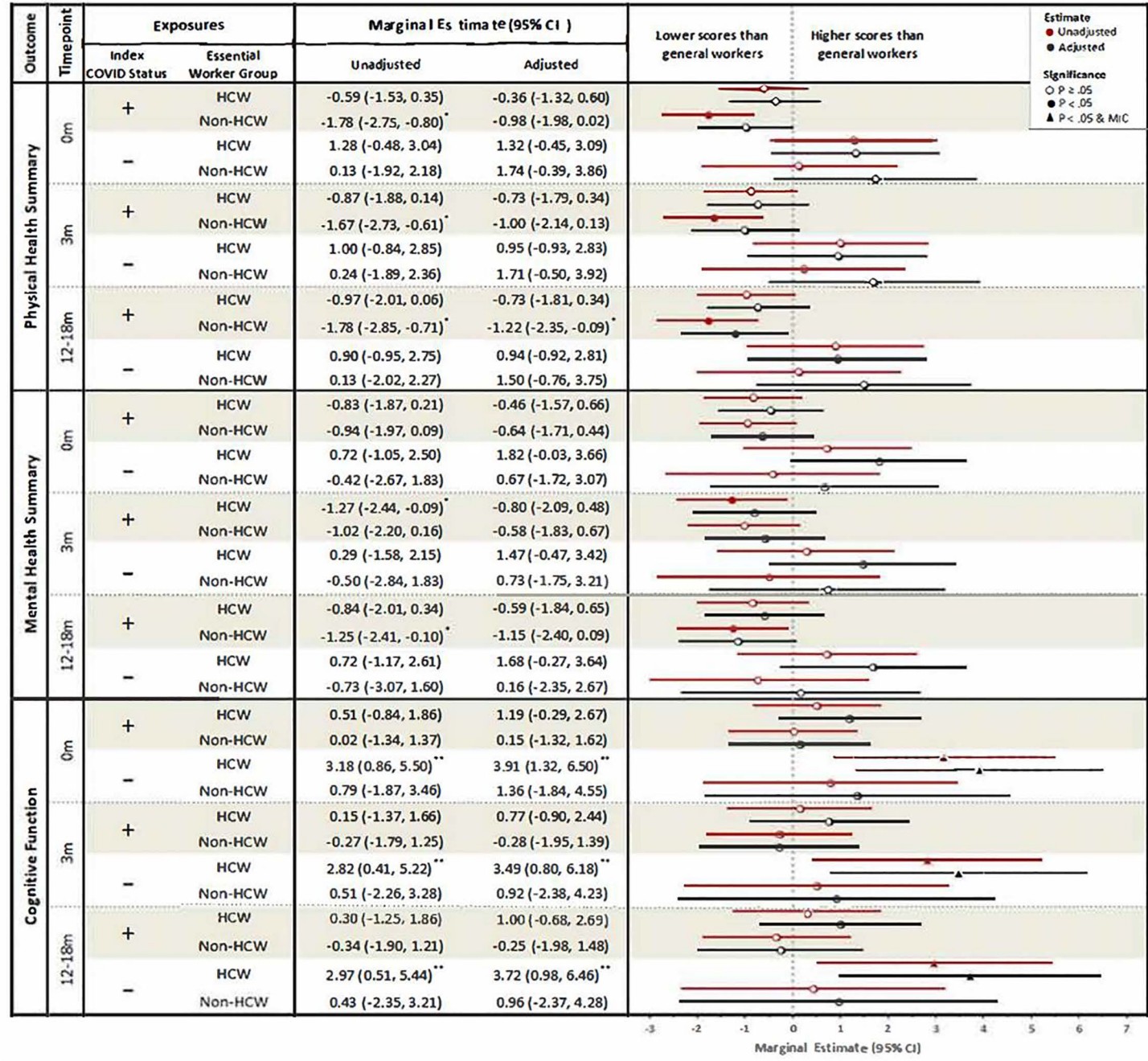

**Fig 2. Marginal estimates of employment type on baseline, short-term, and long-term physical health, mental health, and cognitive function by index COVID (SARS-CoV-2) status.**

Among participants who *did not* have a SARS-CoV-2 infection, essential HCWs had a lower prevalence of poor short-term cognitive function across employment groups and reported cognitive scores that were more than three points higher, on average, than general workers at all timepoints, an association that was both statistically and clinically significant after adjustment. These noteworthy findings could be attributed to several protective factors unique to the healthcare industry

in the context of a global pandemic. Similar to nationally observed trends, essential HCWs in this study were more likely to be younger, have private health insurance, and be vaccinated for SARS-CoV-2 compared to general and essential non-HCW workers.

Furthermore, HCWs undergo extensive professional training, which can increase work resilience and preparedness for emergencies. HCWs are also regularly engaged in mentally challenging activities, which can enhance cognitive resilience [26]. Additionally, their professional environment likely provided more social interaction during the pandemic with built-in support systems and resources for maintaining cognitive health [27]. While these protective characteristics in no way diminish the upheaval and stress inflicted on HCWs during the pandemic, they could help explain differences in health outcomes among different types of essential workers in the absence of a SARS-CoV-2 infection, and highlight the role of occupation in health and well-being during a pandemic. Additionally, it's important to note that the observed protective effect of HCW status on cognitive function was diminished in the context of recovery from SARS-CoV-2 infection. The lack of difference in cognitive function after recovery following SARS-CoV-2 infection may be due to the physiological effects of the coronavirus itself, which may act as an equalizer across employment types. Findings warrant further research to help differentiate the cognitive effects of the COVID-19 pandemic vs. infection across varying occupations.

In contrast to positive findings among essential HCWs, outcomes among essential non-HCWs were less favorable in the context of recovery from SARS-CoV-2 infection. Among COVID-positive participants, essential non-HCWs reported the highest prevalence of observed poor short-term physical health across employment groups at the long-term timepoint. Overall, essential non-HCWs were the most socially vulnerable population, reporting lower levels of education (8% vs. 4.5–5.4% ≤ high school degree) and household income (45.8% vs. 28.7–35.1% < $75,000), a higher prevalence of comorbidities including obesity (34.6% vs. 24.6–24.9%), diabetes (7.8% vs. 2.7–4.6%), and smoking (6.4% vs. 2.5–3.6%), and lower rates of COVID-19 vaccination (59.6% vs. 66.0–74.9%) compared to other groups. However, long-term physical health scores remained 1.22 points lower, on average, among essential non-HCWs compared to general workers after adjusting for participant characteristics.

The association of poor long-term physical health outcomes and essential non-healthcare worker occupation could be secondary to baseline differences in sociodemographics and health status. In this vulnerable population there may also be a contribution from weathering, a conceptual framework that postulates chronic social and economic disadvantage accelerates physical health decline, partly explaining racial disparities in various health conditions [28]. During the COVID-19 pandemic, health disparities among Americans quickly emerged with disproportionately high infection rates and mortality among racial/ethnic minorities and low-income populations [8]. The intersectional relationship of income, race, and employment may be particularly influential in essential non-HCWs, wherein service jobs with inflexible work schedules, stress, lack of professional social support, and limited time off may have hindered physical recovery from COVID-19.

These findings suggest there may be benefit in advancing social safety measures to help reduce health inequities during pandemics. For essential non-HCWs specifically, efforts could be made to improve workplace conditions (e.g., installing air purifiers or filters, allowing flexibility following exposure or outbreaks), promote better health (e.g., reduce or aggressively treat comorbidities, increase COVID-19 vaccination rates through incentives or policy requirements), and increase resiliency for public health emergencies (e.g., increase mental health resources) [29,30].

This study had several strengths, including use of a control group (COVID-negative participants) to enable comparisons by SARS-CoV-2 status, in-person and remote recruitment across geographically dispersed populations and settings, and repeated use of validated PROMIS-29 instruments to accurately capture patient-reported outcomes over 18 months.

The study also has several limitations. First, changes in employment status were not captured in quarterly follow-up surveys, potentially affecting exposures and outcomes. Second, while the study distinguishes between two categories of workers, it does not delve into the occupational heterogeneity within each group, which could have allowed for more precise analyses. This omission limits the ability to identify particularly vulnerable or resilient occupations within each group, which is essential for designing more targeted and effective interventions in the face of future health crises. Third,

requirement of computer or smartphone access for study eligibility may introduce selection bias and limit the generaliz-ability of findings. Fourth, PROMIS-29 outcomes could be influenced by recall and response bias, though surveys were administered in a standardized manner. Fifth, reporting of new SARS-CoV-2 infections during follow-up was not factored into eligibility, potentially skewing outcomes in both COVID groups. Sixth, findings in the COVID-negative group were limited by a smaller sample size and a larger analysis could reveal more meaningful results in this group. Seventh, these findings may not be representative of the working population; participants in this study tended to have fewer social and medical risk factors than people who did not complete follow-up surveys. Eighth, another potential limitation of the study is the lack of consideration for individuals with pre-existing psychopathological conditions within the inclusion or exclusion criteria. This omission may have influenced the findings related to mental health outcomes. Future studies would benefit from accounting for this variable in order to provide a more nuanced interpretation of psychological outcomes and to con-tribute to a more comprehensive understanding of factors influencing mental resilience during public health crises. Lastly, this analysis did not account for the impact of symptom burden, severity and persistence on outcomes over time, which vary across INSPIRE cohorts and warrant future research [18,21].

In conclusion, this study revealed varied differences in patient-reported outcomes among essential workers compared with the general working population following SARS-CoV-2 infection and in the context of the COVID-19 pandemic. While there were significant differences in social determinants and baseline health by worker status, compared with general workers, essential HCWs reported better cognitive function in the absence of SARS-CoV-2 infection at all timepoints. Essential non-HCWs reported worse long-term physical health than general workers following SARS-CoV-2 infection, although the difference was mitigated after adjusting for social risk factors and comorbidities . Preparation efforts for future pandemics should prioritize enhanced protection and post-infection resources for frontline workers.

## Supporting information

**S1 Appendix.** Appendix 1: Acknowledgements INSPIRE Group Author List. Appendix 2: Demonstration of Adjusted Effect Estimation from GEE Modeling.
(DOCX)

## Acknowledgments

The Innovative Support for Patients with SARS-CoV-2 Infections Registry (INSPIRE) is funded by the Centers for Disease Control and Prevention (CDC, www.cdc.gov), National Center of Immunization and Respiratory Diseases (NCIRD) (contract number: 75D30120C08008; PI, Robert A. Weinstein, MD). The findings and conclusions in this report are those of the author(s) and do not necessarily represent the official position of the Centers for Disease Control and Prevention (CDC). We would like to thank California Department of Public Health for their assistance with participant recruitment for this study, as well as the CTSI COVID Clinical Research Steering Committee and the CTSI Office of Clinical Research Patient Navigation Team and Bioinformatics Program for assistance with study recruitment. NIH/NCATS #UL1TR001881.

## Author contributions

**Conceptualization:** Jocelyn Dorney, Caitlin Malicki, Lauren E. Wisk, Joann Elmore, Kelli N. O'Laughlin, Dana Morse, Michael A Gottlieb, Michelle Santangelo, Sharon Saydah, Robert A Weinstein, Erica S Spatz.

**Data curation:** Imtiaz Ebna Mannan.

**Formal analysis:** Imtiaz Ebna Manan.

**Investigation:** Jocelyn Dorney, Imtiaz Ebna Mannan, Caitlin Malicki, Lauren E. Wisk, Joann Elmore, Michael A Gottlieb, Robert A Weinstein, Erica S Spatz.

**Methodology:** Jocelyn Dorney, Caitlin Malicki, Michael A Gottlieb, Robert A Weinstein, Erica S Spatz, Lauren E Wisk, Imtiaz Ebna Mannan.

**Project administration:** Jocelyn Dorney, Michael A Gottlieb, Michelle Santangelo.

**Resources:** Robert A Weinstein, Erica S Spatz.

**Supervision:** Erica S Spatz.

**Writing – original draft:** Jocelyn Dorney, Imtiaz Ebna Mannan, Caitlin Malicki, Lauren E. Wisk, Joann Elmore, Kelli N. O'Laughlin, Dana Morse, Kristyn Gatling, Michael A Gottlieb, Michelle Santangelo, Nicole L. Gentile, Michelle L'Hommedieu, Sharon Saydah, Ryan Huebinger, Katherine Riley Martin, Ahamed H Idris, Efrat Kean, Kevin Schaeffer, Robert M Rodriguez, Robert A Weinstein, Erica S Spatz.

**Writing – review & editing:** Jocelyn Dorney, Imtiaz Ebna Mannan, Caitlin Malicki, Lauren E. Wisk, Joann Elmore, Kelli N. O'Laughlin, Dana Morse, Kristyn Gatling, Michael A Gottlieb, Michelle Santangelo, Nicole L. Gentile, Michelle L'Hommedieu, Sharon Saydah, Mandy J. Hill, Ryan Huebinger, Katherine Riley Martin, Ahamed H Idris, Efrat Kean, Kevin Schaeffer, Robert M Rodriguez, Robert A Weinstein, Erica S Spatz.

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
