## [Decision Letter · Decision Letter 0]

Dear Dr. Spatz,

Thank you for submitting your manuscript to PLOS ONE. After careful consideration, we feel that it has merit but does not fully meet PLOS ONE’s publication criteria as it currently stands. Therefore, we invite you to submit a revised version of the manuscript that addresses the points raised during the review process.

Dear Authors, this manuscript is in need of a minor revision before it's ready for next steps. Please kindly address our reviewers' concerns to enhance the quality of your work. Thank you very much!

We look forward to receiving your revised manuscript.

Kind regards,

Harapan Harapan, MD, PhD

Academic Editor

PLOS ONE

2. Thank you for stating the following financial disclosure:  [The Innovative Support for Patients with SARS-COV-2 Infections Registry (INSPIRE) is funded by the Centers for Disease Control and Prevention (CDC, www.cdc.gov), National Center of Immunization and Respiratory Diseases (NCIRD) (contract number: 75D30120C08008; PI, Robert A. Weinstein, MD).].  Please state what role the funders took in the study.  If the funders had no role, please state: "The funders had no role in study design, data collection and analysis, decision to publish, or preparation of the manuscript." If this statement is not correct you must amend it as needed. Please include this amended Role of Funder statement in your cover letter; we will change the online submission form on your behalf.

Reviewers' comments:

Reviewer's Responses to Questions

**Comments to the Author**

1. Is the manuscript technically sound, and do the data support the conclusions?

Reviewer #1: Yes

Reviewer #2: Partly

Reviewer #3: Yes

2. Has the statistical analysis been performed appropriately and rigorously?

Reviewer #1: Yes

Reviewer #2: Yes

Reviewer #3: Yes

3. Have the authors made all data underlying the findings in their manuscript fully available?

Reviewer #1: Yes

Reviewer #2: Yes

Reviewer #3: Yes

4. Is the manuscript presented in an intelligible fashion and written in standard English?

Reviewer #1: Yes

Reviewer #2: Yes

Reviewer #3: Yes

Reviewer #1: This ia an important article to address the patient-reported outcomes among essential workers compared with the general working population following SARS-CoV-2 infection and in the context of the COVID-19 pandemic.. I recommended acceptance for publication. The only interested further analysis is whether the patient-reported outcomes are different among different SARS-CoV2 variants.

Reviewer #2: A review of the manuscript entitled “Impact of SARS-CoV-2 on Healthcare and Essential Workers: A Longitudinal Study of PROMIS-29 Outcomes”

1. The authors have clearly written this manuscript in professional, unambiguous language. I appreciate the authors for their extensive data set, which included participants from eight US sites. However, there are some minor things that need to be rectified to increase the quality of this study.

2. (page 16) please revise the study timing from this sentence using a unified format: “A total of 8950 participants enrolled in the study between 12/7/2020 and 8/29/22, of which 6,044 were eligible for follow-up based on the three sets of eligibility criteria above.”

3. Authors are suggested to proofread the manuscript after addressing all comments to avoid any typological, grammatical, and lingual mistakes and errors. For example, “Each site began their enrollment” on page 16; “…after (omicron) this period.” on page 20; “Among essential non-HCWs…” on page 24.

4. Before delving further into COVID-19 in the US, I suggest that authors elaborate more on what COVID-19 itself is first. For example, “The COVID-19 pandemic has profoundly impacted millions of lives worldwide. By June 2023, the global count of confirmed COVID-19 cases exceeded 767 million, with over 6.9 million fatalities. Healthcare workers have borne the brunt of the pandemic's effects, with the World Health Organization (WHO) reporting that over 14% of COVID-19 cases involve this group. The rising incidence of COVID-19 among healthcare professionals risks significantly straining healthcare systems, reducing service quality, and leading to adverse health outcomes.” This information can be cited from “Fatigue in healthcare workers with mild COVID-19 survivors in Indonesia”.

5. (page 14, introduction) This sentence “Frontline workers risked their personal and household’s health and well-being, with ongoing exposure to severe acute respiratory syndrome coronavirus 2 (SARS-CoV-2)…” also shares the same idea with a similar study by Hamdan et al. entitled “Coping strategies used by healthcare professionals during COVID-19 pandemic in Dubai: A descriptive cross-sectional study”. Kindly include this reference to make the sentence stronger.

6. (page 15, methods) I suggest that authors arrange the information on the start dates for each institution in a table to make it more organised.

Reviewer #3: Dear Authors,

Thank you for this article. It is an exciting article to publish. The title is Impact of SARS-CoV-2 on Healthcare and Essential Workers: A Longitudinal Study of PROMIS-29 Outcomes, and the ID number is (PONE-D-24-50826). The findings can be compared to general workers; essential HCW status had higher cognitive function in the absence of SARS-CoV-2 infection at all time points, while essential non-HCWs were most vulnerable to poor recovery in long-term physical health following SARS-CoV-2 infection. Preparation efforts for future pandemics may consider enhanced protection and post-infection resources for frontline workers.

- The results in this article are good.

Finally, this article is acceptable for publishing because it is the new result and well written.

**Do you want your identity to be public for this peer review?** For information about this choice, including consent withdrawal, please see our Privacy Policy

Reviewer #1: **Yes: ** Tun-Chieh Chen

Reviewer #2: No

Reviewer #3: No

---

## [Author Response · Author response to Decision Letter 1]

12 Mar 2025

Thank you for the opportunity to revise this manuscript. We appreciate the Editors and Reviewers comments and have addressed them below and in the manuscript.

Reviewer #1: This is an important article to address the patient-reported outcomes among essential workers compared with the general working population following SARS-CoV-2 infection and in the context of the COVID-19 pandemic. I recommended acceptance for publication. The only interested further analysis is whether the patient-reported outcomes are different among different SARS-CoV2 variants.

Thank you for your comments and recommendation that this manuscript be accepted for publication. Your recommendation for further analysis is warranted and noted.

Reviewer #2: A review of the manuscript entitled “Impact of SARS-CoV-2 on Healthcare and Essential Workers: A Longitudinal Study of PROMIS-29 Outcomes”

1. The authors have clearly written this manuscript in professional, unambiguous language. I appreciate the authors for their extensive data set, which included participants from eight US sites. However, there are some minor things that need to be rectified to increase the quality of this study.

Thank you for your comments and suggestions.

2. (page 16) please revise the study timing from this sentence using a unified format: “A total of 8950 participants enrolled in the study between 12/7/2020 and 8/29/22, of which 6,044 were eligible for follow-up based on the three sets of eligibility criteria above.”

This has been corrected to state: “A total of 8,950 participants enrolled in the study between 12/7/2020 and 8/29/2022, of which 6,044 were eligible for follow-up based on the three sets of eligibility criteria above.”

3. Authors are suggested to proofread the manuscript after addressing all comments to avoid any typological, grammatical, and lingual mistakes and errors. For example, “Each site began their enrollment” on page 16;

Thank you for noticing this error, this clause has been deleted.

“…after (omicron) this period.”

Thank you for highlighting this error, it was corrected to state “19% enrolled during the omicron variant period”

on page 20; “Among essential non-HCWs…” on page 24.

Thank you for highlighting this grammatical error, among has been changed to “For”

4. Before delving further into COVID-19 in the US, I suggest that authors elaborate more on what COVID-19 itself is first. For example, “The COVID-19 pandemic has profoundly impacted millions of lives worldwide. By June 2023, the global count of confirmed COVID-19 cases exceeded 767 million, with over 6.9 million fatalities. Healthcare workers have borne the brunt of the pandemic's effects, with the World Health Organization (WHO) reporting that over 14% of COVID-19 cases involve this group. The rising incidence of COVID-19 among healthcare professionals risks significantly straining healthcare systems, reducing service quality, and leading to adverse health outcomes.” This information can be cited from “Fatigue in healthcare workers with mild COVID-19 survivors in Indonesia”.

This reference has been added and the suggested text was condensed to: “The COVID-19 pandemic has profoundly impacted millions of lives worldwide. By June 2023, the global count of confirmed COVID-19 cases exceeded 767 million, with over 6.9 million fatalities. Healthcare workers have borne the brunt of the pandemic's effects, with the World Health Organization (WHO) reporting that over 14% of COVID-19 cases involve this group [1].”

5. (page 14, introduction) This sentence “Frontline workers risked their personal and household’s health and well-being, with ongoing exposure to severe acute respiratory syndrome coronavirus 2 (SARS-CoV-2)…” also shares the same idea with a similar study by Hamdan et al. entitled “Coping strategies used by healthcare professionals during COVID-19 pandemic in Dubai: A descriptive cross-sectional study”. Kindly include this reference to make the sentence stronger.

Thank you for sharing more supporting information, this paper has been added as reference 5 in the manuscript.

6. (page 15, methods) I suggest that authors arrange the information on the start dates for each institution in a table to make it more organised.

Thank you for your suggestion to increase organization, I have created a new table 1 to organize this information appropriately.

Table 1: Institutions start date of enrollment in the INSPIRE Study

Institution Start Date

Rush University November 17, 2020

University of Washington December 11, 2020

Yale University December 26, 2020

University of California, Los Angeles February 1, 2021

Thomas Jefferson University February 8, 2021

University of California, San Francisco (UCSF) February 24, 2021

University of Texas Southwestern (UTSW) April 21, 2021

University of Texas Health Science Center at Houston (UTH) May 5, 2021

Reviewer #3: Dear Authors,

Thank you for this article. It is an exciting article to publish. The title is Impact of SARS-CoV-2 on Healthcare and Essential Workers: A Longitudinal Study of PROMIS-29 Outcomes, and the ID number is (PONE-D-24-50826). The findings can be compared to general workers; essential HCW status had higher cognitive function in the absence of SARS-CoV-2 infection at all time points, while essential non-HCWs were most vulnerable to poor recovery in long-term physical health following SARS-CoV-2 infection. Preparation efforts for future pandemics may consider enhanced protection and post-infection resources for frontline workers.

- The results in this article are good.

Finally, this article is acceptable for publishing because it is the new result and well written.

Thank you for your comments and recommendation for this manuscript to be published.

---

## [Decision Letter · Decision Letter 1]

Impact of SARS-CoV-2 on Healthcare and Essential Workers: A Longitudinal Study of PROMIS-29 Outcomes

PONE-D-24-50826R1

Dear Dr. Spatz,

We’re pleased to inform you that your manuscript has been judged scientifically suitable for publication and will be formally accepted for publication once it meets all outstanding technical requirements.

Kind regards,

Harapan Harapan, MD, PhD

Academic Editor

PLOS ONE

Additional Editor Comments (optional):

Reviewers' comments:

Reviewer's Responses to Questions

**Comments to the Author**

Reviewer #2: All comments have been addressed

Reviewer #4: (No Response)

2. Is the manuscript technically sound, and do the data support the conclusions?

Reviewer #2: Yes

Reviewer #4: Yes

3. Has the statistical analysis been performed appropriately and rigorously?

Reviewer #2: Yes

Reviewer #4: Yes

4. Have the authors made all data underlying the findings in their manuscript fully available?

Reviewer #2: Yes

Reviewer #4: Yes

5. Is the manuscript presented in an intelligible fashion and written in standard English?

Reviewer #2: Yes

Reviewer #4: Yes

Reviewer #2: I commend the authors for addressing my concerns. Now the manuscript has improved a lot and is ready for further consideration.

Reviewer #4: The review of this study has been highly enriching, as it constitutes a work of notable scientific relevance. I consider its publication to be fully justified, as it addresses a medical and social issue with direct implications for the quality of life of a population subgroup that requires specific and comprehensive care. With the aim of contributing to its proper presentation, I offer the following observations:

- It is recommended to standardize the format of the bibliographic references, as inconsistencies have been observed in various sections of the article regarding the spacing before brackets ([#]).

- One potential limitation of the study is the lack of consideration for individuals with pre-existing psychopathological conditions within the inclusion or exclusion criteria. This omission may have influenced the findings related to mental health outcomes. As highlighted by Manchia et al. (2021) in their review "The impact of the prolonged COVID-19 pandemic on stress resilience and mental health: A critical review across waves", such conditions may predispose individuals to internalizing symptoms. Future studies would benefit from accounting for this variable in order to provide a more nuanced interpretation of psychological outcomes and to contribute to a more comprehensive understanding of factors influencing mental resilience during public health crises.

- While the study distinguishes between two categories of workers, it does not delve into the occupational heterogeneity within each group, which could have allowed for more precise analyses. This omission limits the ability to identify particularly vulnerable or resilient occupations within each group, which is essential for designing more targeted and effective interventions in the face of future health crises.

I would like to thank the editor for the opportunity to review this important study. I believe it is essential to continue investigating the effects of the COVID-19 pandemic, not only on health but across all aspects of human life. Studies like this provide valuable information that can help guide strategies to more effectively address future public health crises.

**Do you want your identity to be public for this peer review?** For information about this choice, including consent withdrawal, please see our Privacy Policy

Reviewer #2: No

Reviewer #4: **Yes: ** Araceli Guerra Martínez

---

## [Editor Report · Acceptance letter]

PONE-D-24-50826R1

PLOS ONE

Dear Dr. Spatz,

I'm pleased to inform you that your manuscript has been deemed suitable for publication in PLOS ONE. Congratulations! Your manuscript is now being handed over to our production team.

Kind regards,

on behalf of

Dr. Harapan Harapan

Academic Editor

PLOS ONE